# Delivery of RNAs to Specific Organs by Lipid Nanoparticles for Gene Therapy

**DOI:** 10.3390/pharmaceutics14102129

**Published:** 2022-10-07

**Authors:** Kelly Godbout, Jacques P. Tremblay

**Affiliations:** 1Centre de Recherche du CHU de Québec, Laval University, Quebec, QC G1V 4G2, Canada; 2Department of Molecular Medicine, Faculty of Medicine, Laval University, Quebec, QC G1V 0A6, Canada

**Keywords:** lipid nanoparticles, gene therapy, mRNA delivery, CRISPR/Cas9 delivery, specific organ delivery

## Abstract

Gene therapy holds great promise in the treatment of genetic diseases. It is now possible to make DNA modifications using the CRISPR system. However, a major problem remains: the delivery of these CRISPR-derived technologies to specific organs. Lipid nanoparticles (LNPs) have emerged as a very promising delivery method. However, when delivering LNPs intravenously, most of the cargo is trapped by the liver. Alternatively, injecting them directly into organs, such as the brain, requires more invasive procedures. Therefore, developing more specific LNPs is crucial for their future clinical use. Modifying the composition of the lipids in the LNPs allows more specific deliveries of the LNPs to some organs. In this review, we have identified the most effective compositions and proportions of lipids for LNPs to target specific organs, such as the brain, lungs, muscles, heart, liver, spleen, and bones.

## 1. Introduction

Over the last few years, the enthusiasm for gene therapy [1] has brought out the best in science. Whether through the CRISPR system [2,3,4,5,6] or through RNAs [7], DNA modifications have now become possible and hold great promise. However, a critical question remains unsolved—how to deliver these technologies to specific organs. Several delivery methods have been attempted, such as adeno-associated viruses (AAVs) [8,9,10], extracellular vesicles (EVs) [11,12], lipid nanoparticles (LNPs) [13,14,15,16,17,18,19], etc. Since they were used in Pfizer and Moderna’s COVID-19 vaccines, LNPs are in the spotlight as their safety and effectiveness have been widely proven [20,21]. Furthermore, one of the great advantages of LNPs is their large capacity. They can package large components such as long RNAs and large proteins [22]. This has paved the way for the delivery of the CRISPR technology. This is where LNPs have taken the lead over AAVs, as the capacity of AAVs is at most 5 kb [23]. Regardless of the delivery method, one problem is still present. When delivered intravenously (IV), most of the LNPs are taken up by the liver [24]. Moreover, injecting them directly into an organ, such as the brain, requires invasive techniques [25]. Therefore, modifications of the composition of LNPs to make them more specific are needed. In order to be optimal and functional, LNPs must contain certain classes of lipids.

LNPs are generally composed of four categories of lipids: ionizable cationic lipids, helper lipids (phospholipids), cholesterol, and polyethylene glycol (PEG) lipids (Figure 1) [26,27,28,29]. Each of these lipids plays an important role [24,30]. Ionizable cationic lipids have a positive charge that allows them to interact with nucleic acids (which are negatively charged) [31]. When included in LNPs, these lipids allow nucleic acids to be loaded into the particles. A crucial feature of these lipids is that they become protonated at acidic pH. When they become positively charged, the membrane of the particle is destabilized. This allows the LNPs to escape from the endosomes and release their cargo into the cytosol of the cells [32]. These lipids also enhance the efficiency of endocytosis by target cells. Helper lipids are mostly phospholipids. They improve the stability of the nanoparticle [33]. They also increase the efficiency of the delivery [34]. Cholesterol is an essential lipid to include in LNPs. It fills the gaps between the phospholipids, and this increases the stability of the particles [27,34]. PEG lipids constitute the last classical part added to the LNP formulation. These lipids greatly increase the circulation time of LNPs in the body [27,35]. They also decrease immunogenicity [36].

By manipulating the composition of lipids (Figure 2) and their proportion in LNPs, it is possible to change their size and surface characteristics. Indeed, the size and charge of LNPs seem to have significant effects on their biodistribution. Therefore, the delivery of the LNPs can be more specific to certain organs. Unfortunately, we still do not clearly understand the precise mechanism that explains the relationship between the biodistribution of LNPs and their size, charge, and the types of lipids used. It would therefore be interesting to conduct fundamental research on this mechanism. Although the explanation behind the biodistribution of some LNPs is not fully understood, the scientific literature is replete with several formulations of LNPs having the ability to target a specific organ. In this review, we have identified the most effective formulations of lipid nanoparticles for passive targeting of specific organs, such as muscles, brain, lungs, liver, heart, spleen, and bones. Active targeting is another strategy to target specific organs, but it will not be discussed in this review. For more information on this strategy, Menon et al. [37] have published an article on this subject. As passive targeting with lipid nanoparticles is a vast subject in itself, in this review we will discuss the types of lipids to use, as well as their proportions, in order to deliver gene therapies to specific organs, especially extrahepatic ones.

## 2. Targeting the Muscles

Myopathies (e.g., Duchenne’s muscular dystrophy (DMD) [55,56], Steinert’s myopathy [57], diseases related to a mutation in the RYR1 gene [58], and many others [59]) greatly affect the quality of life and the life expectancy of many patients. To eventually treat these patients, many researchers have developed drugs [60] or used gene therapy [61,62]. However, it is still very difficult to deliver the therapy specifically to the muscles of the whole body.

Different routes have been used in attempts to target the muscles (Table 1). Intramuscular injections have been attempted [63,64,65,66]. Some researchers have also tried to reach the muscles via intravenous [63,67,68] or intraperitoneal injection [69].

Only a few studies have succeeded in targeting muscles through intravenous injections of LNPs. The major problem with this route is that LNPs are taken up by the liver and the spleen [70]. Kenjo et al. [63] delivered CRISPR-Cas9 mRNA with a sgRNA targeting the *DMD* gene by injecting LNPs in the dorsal saphenous vein. They used the following composition: TCL053/DPPC/cholesterol/DMG-PEG (60:10.6:27.3:2.1). The lipid/RNA weight ratio was 23:1. The size of these LNPs was 79.1 nm. When the injection volume of the limb perfusion was greater than 5 mL/kg, they succeeded in editing the target gene in various limb muscles. However, it is important to note that they used a tourniquet at the bottom of the quadriceps. This method allows the LNPs to remain concentrated around the muscles under the tourniquet without traveling to the liver or other organs. However, tourniquets can be used for the legs and the arms but are useless for the other muscles of the body, such as the diaphragm.

Kenjo et al. [63] also tried the same lipid composition that was used intravenously for intramuscular injection in the tibialis anterior (TA) muscle of a mouse model of Duchenne muscular dystrophy (DMD). After six injections, dystrophin expression was restored in 38.5% of muscle fibers. In each injection, the authors used 10 μg of Cas9 mRNA with 10 μg of sgRNA targeting the *DMD* gene to provide exon skipping. As six is a high number of injections, some modifications should be identified to increase the method’s efficiency at a lower number of injections. For comparison, after three injections, they obtained only about 15% exon skipping and around 4.0% dystrophin recovery.

Wei et al. [66] used an interesting composition, with which they succeeded in encapsulating ribonucleoprotein complexes (RNPs, i.e., Cas9 proteins and sgRNAs) and delivering them to the muscle via intramuscular injections. RNPs are interesting because the Cas9 nuclease does not need to be translated since the protein is already there. Compared to Cas9 mRNA delivery, direct delivery of the Cas9 protein makes editing much faster and more direct. This strategy has also been shown to decrease the risk of off-target effects [71]. By avoiding a sustained expression of Cas9, this strategy has low immunogenicity [72,73,74,75,76,77,78,79].

However, the delivery of the Cas9 protein is much more challenging than the delivery of its mRNA. The large size of this protein makes it more difficult to encapsulate. It is also more complicated to avoid degradation or denaturation of the protein during the assembly of the LNPs and the delivery process. Since sgRNAs have a highly negative charge, they are more difficult to encapsulate in LNPs [24].

Some lipid compositions of LNPs are therefore more useful for delivering RNPs. By adding a permanently charged cationic lipid, such as DOTAP, to a conventional LNP composition, Wei et al. [66] were able to preserve the integrity of the RNPs and deliver them to their destinations.

The molar ratio of Cas9/sgRNA used was 3/1. The sgRNA was targeting the *DMD* gene. Their LNPs were composed of 5A2-SC8/DOPE/cholesterol/DMG-PEG/DOTAP = 21.4/21.4/42.8/4.3/10 (mol/mol) and their lipid/RNA ratio was 40:1 (w:w) [66]. Their composition was innovative because it included, in addition to the four usual lipids, DOTAP, which is a permanently charged cationic lipid. They found that when using a ratio of 10% of DOTAP, the composition was more specific for muscles. After three injections, they obtained a 4.2% restoration of dystrophin [66]. They also demonstrated that adjusting the concentration of DOTAP changed the specificity of LNPs and targeted different organs (see sections on other organs).

Carrasco et al. [65] delivered LNPs containing FLuc mRNA via intramuscular injections. Their LNPs were composed of DLin-KC2-DMA/DSPC/cholesterol/DMG-PEG (50/10/38.5/1.5), with a 4:1 mole/mole ratio of ionizable lipid:mRNA. Their results indicated that the LNPs reached the entire leg of the mouse, as well as (to a lower extent) the liver. However, they did not present quantitative results. It would therefore be interesting to test their composition with Cas9 mRNA and an sgRNA targeting the *DMD* gene, and to determine the percentage of editing in all the different muscles of the leg.

Guimaraes et al. [67] created a library of LNPs containing custom-designed barcoded mRNAs (b-mRNAs). They injected these LNPs into mice intravenously. Using this approach, they identified LNP formulations that could be used to target specific organs. Among the studied LNP compositions, two of them stood out as being effectively delivered to the muscles (the formulations were named F11 and F12). Their composition was as follows: C12-200/DOPE/cholesterol/DMG-PEG with a ratio of 35/16/46.5/2.5. F11 and F12 differed only in terms of their ionizable lipid:mRNA ratios, and therefore their size. F11 had a ratio of 5:1 and a size of 83.55 nm and F12 had a ratio of 7.5:1 and a size of 88.09 nm [67]. Since we do not have the raw data on the number of reads per organ for each LNP, we cannot conclude anything about the specificity of these LNPs. Their specificity would therefore be interesting to evaluate. However, we do know that these LNPs can efficiently reach the muscles following an intravenous injection, which is very relevant to the goal of reaching all the muscles of the body.

Based on all these articles, it is clear that the muscles remain very difficult to reach via IV methods without the LNPs also becoming trapped by other tissues. However, some points stand out when comparing all of these studies. The only one that has so far succeeded in delivering CRISPR-Cas9 mRNA and a sgRNA into muscle intravenously is that of Kenjo et al. [63]. However, they used tourniquets on the mouse legs, which is not realistic to use in humans for all muscles. Therefore, to target all muscles intravenously, it would be interesting to experiment with the encapsulation of CRISPR-Cas9 (mRNA or RNP) with an sgRNA in one of the LNPs described by Guimaraes [67]. It should be noted that in both studies, the ionizable cationic lipid was C12-200. This suggests that this ionizable cationic lipid may increase the specificity of LNPs for the muscles.

## 3. Targeting the Brain

The brain is the center of many diseases that are still not curable. Alzheimer’s disease [80], Huntington’s chorea [81], epilepsy [82], and brain cancer [83] are just some examples of brain diseases. It is becoming urgent to find a safe and effective method to deliver treatments to the brain.

The brain is very hard to target. It is protected by the skull and the blood–brain barrier (BBB) [84,85,86], which is very selective for brain penetration. Some investigators have used drastic methods such as intracranial injections [87]. However, this procedure involves huge risks and is very invasive. Since we aimed to identify a minimally invasive and safe delivery method, we discuss the experiments that have reached the brain via intravenous injections in greater detail (Table 2).

Ma et al. [88] developed a class of neurotransmitter-derived lipidoids (NT-lipidoids). These researchers are innovative because they are among the few who have attempted to add neurotransmitter-derived lipidoids (NT-lipidoids) to prior BBB-impermeable LNPs. With the NT-lipidoids allowing them to cross the BBB, these LNPs have successfully delivered antisense oligonucleotides (ASOs) against tau and the genome-editing fusion protein (-27)GFP-Cre recombinase into the mouse brain via systemic intravenous administration.

To deliver ASOs against tau, they used the following composition: 306-O12B-3/DSPE-PEG/NT1-O14B with a respective ratio of 67.2/4/28.8 (*w*/*w*). They had a total lipid:ASO ratio of 15:1. This led to an approximately 50% reduction in tau mRNA and an approximately 30% reduction in tau protein. Here, the tail vein intravenous method was even more efficient than the implanted ICV pump. To deliver the genome-editing fusion protein (−27)GFP-Cre recombinase, they injected into Ai14 mice the following formulation: PBA-Q76-O16B/DSPE-PEG/NT1-O14B with a respective ratio of 67.2/4/28.8 (*w*/*w*). This experiment was also successful. They observed strong tdTomato signals in multiple regions of the brain, including the cerebral cortex, hippocampus, and cerebellum. These results are therefore very encouraging. It would, however, be interesting to test the specificity of these LNPs to check that they do not travel to the liver or other organs.

Nabhan et al. [89] did not directly target the brain but rather the DRG (dorsal root ganglia). They intrathecally injected LNPs containing the mRNA of the human *FXN* gene into BALB/c mice to perform RNA transcript therapy (RTT). The composition of their LNPs was as follows: DLin-MC3-DMA/DSPC/cholesterol/DMG-PEG, with a respective ratio of 55/10/32.5/2.5 and a lipid/mRNA weight ratio of 30:1 (w:w). Their LNPs had a size of 85 nm. Their results revealed that human mRNA *FXN* levels in the DRG were about three-fold higher than mouse mRNA *FXN* in the control group. Thus, they were able to observe an increase in the level of frataxin protein in the DRG.

They also tried to inject these LNPs into CD1-mice via IV [89]. However, these LNPs were significantly taken up by the liver, but not by the heart and brain, which are the organs affected by ataxia. Fortunately, the additional amount of mFXN in the liver did not adversely influence other mitochondrial proteins.

Therefore, for treatments for ataxia or other diseases affecting the DRG, it would be better to inject the LNPs described above intrathecally [89]. If the brain is to be targeted, it would be judicious to try the formulations described above by Ma et al. [88].

## 4. Targeting the Lungs

The lungs are the site of devastating diseases. Cystic fibrosis [91,92] and lung cancer [93] come to mind. Fortunately, the delivery of drugs to the lungs is one of the most studied topics in this field (Table 3). For treatments that must reach the lungs, three routes have been prioritized: intravenous, intranasal, and inhalation.

Cheng et al. [94] developed a selective organ targeting (SORT) strategy. They found that the addition of a SORT molecule, 1,2-dioleoyl-3-trimethylammonium-propane (DOTAP), to the LNPs permitted them to target an organ specifically. These LNPs are also interesting because they allowed one to deliver multiple gene editing technologies, including mRNA, Cas9 mRNA/single-guide RNA, and Cas9 ribonucleoprotein complexes. To target the lungs, they found that the best LNP had 50% of DOTAP. It had the following formulation: 5A2-SC8/DOPE/Chol/DMG-PEG/DOTAP (11.9/11.9/23.8/2.4/50). First, they intravenously injected these LNPs containing Cas9 mRNA/sgRNA in a 4:1 ratio (w:w). The size of these LNPs was 113.1 nm. They succeeded in editing PTEN exclusively in the lungs with 15.1% indels. Secondly, they intravenously injected these LNPs containing the Cas9 protein with an sgRNA in a 2:1 ratio (w:w). They succeeded in editing specifically in the lungs at 5.3%. They also evaluated the off-target effects for PTEN. Fortunately, they did not detect any off-target DNA editing.

Wei et al. [66] succeeded in encapsulating RNPs and delivering them to the lungs via tail vein injection. The mole ratio of Cas9/sgRNA used was 3:1. Their LNPs were composed of 5A2-SC8/DOPE/cholesterol/DMG-PEG/DOTAP at 11.9/11.9/23.8/2.4/50 (mol/mol) and their lipid/RNA ratio was 40:1 (w:w). As Cheng et al. noted, their composition was innovative as it used DOTAP in addition to the four usual lipids. They noted that when using a ratio of 50% DOTAP, the composition was more specific for the lungs. They intravenously injected Td-Tom mice with these LNPs containing the Cas9 protein and six different sgRNAs (sgTOM, sgP53, sgPTEN, sgEml4, sgALK, and sgRB1). They succeeded in editing the *TOM* gene, detecting bright Td-Tom fluorescence in the lungs after one week. They also succeeded in editing five other genes in the lungs. They observed clear T7EI cleavage bands. P53, PTEN, EMI4, ALK, and RB1 had 1.1%, 3.4%, 7.7%, 1.1%, and 7.5% of indels, respectively. Their experiments demonstrated that these LNPs were able to target the lungs specifically and to edit multiple genes in this organ effectively at low doses (0.33 mg/kg for each sgRNA).

Liu et al. [101] also obtained impressive results. They conducted three LNP delivery experiments, each one containing a different package. In the first one, they inserted Cre mRNA into the LNPs and injected them intravenously into Cre-LoxP mouse models. This LNP succeeded in transfecting about 34% of all endothelial cells, ~20% of all epithelial cells, and ~13% of immune cells. Their LNP formulation contained 9A1P9/DDAB/cholesterol/DMG-PEG (46/23/30.7/0.3 molar ratio). Their ionizable lipid:mRNA ratio was 18:1. The particle size was around 150 nm. They used this same LNP formulation to send the Cas9 mRNA and Tom1 sgRNA intravenously. This experiment resulted in specific gene editing of the lungs. In their final experiment, they inserted Cas9 mRNA and a sgRNA for PTEN with these same LNPs. They obtained efficient target gene editing.

Hagino et al. [102] made a unique LNP. They created a double-coated LNP decorated with the GALA peptide. The GALA peptide has a high affinity for the lung endothelium [103] and was therefore used as a ligand to target the lung endothelium. This peptide was also useful as an endosomal escape device. The inner coat (half of the total lipid) was made of DOPE/STR-R8 (9.55/0.45). The outer coat (the other half) was made of DOTMA/YSK05/cholesterol/DMG-PEG/chol-GALA (4/4/2/0.3/0.4). The size of their LNPs was 125-155 nm. Hagino et al. reported that with this double-coated LNP with the GALA peptide, they obtained one of the highest lung selectivity values reported to date. They estimated the amount of luciferase protein in the lung tissues at ∼74 ng/mg of protein. When delivering the pDNA/PEI complex with this LNP, they observed high gene expression in the lungs. It would be interesting to test the encapsulation of the Cas9 mRNA or protein and the sgRNA with this formulation to verify whether the specificity of these LNPs is preserved.

## 5. Targeting the Liver

Many diseases can affect the liver. Among them, tyrosinemia [104,105] and autoimmune hepatitis [106] are very severe diseases. Fortunately, LNP delivery to the liver is very effective. It is enhanced by the biological effect of apolipoprotein E (ApoE) [107,108]. After LNPs are injected intravenously, ApoE forms a corona around the particles. Since ApoE binds specifically to receptors on hepatocytes, it directs LNPs towards the liver. This explains why LNPs are more easily trapped by the liver [108]. In addition, the liver can be used as a protein factory [109] to treat various rare diseases through gene therapy. As the liver is an excellent therapeutic target, it is important to optimize delivery to this organ.

Patisiran (ONPATTRO™) is the first RNA drug delivered by LNPs that has been approved by the U.S. Food and Drug Administration (FDA) [107]. This LNP is composed of DLin-MC3-DMA/DSPC/cholesterol/DMG-PEG at a molar ratio of 50:10:38.5:1.5 [110]. As it was the first to be approved, this formulation inspired the composition of other LNPs. Table 4 presents some formulations that target the liver.

Liu et al. [22] obtained impressive results. When delivering LNPs with Cas9 mRNA, along with a sgRNA targeting PCSK9, they succeeded in reducing the serum PCSK9 by 80%. They used the following formulation: BAMEA-16B/DOPE/cholesterol/DSPE-PEG (16/4/8/1 w:w). They used a lipid:mRNA ratio of 15:1 (w:w). Their LNP had a size of 230 nm. Fortunately, they observed no signs of inflammation and no obvious hepatocellular injury.

Rothgangl et al. [113] have obtained important results. They tested the efficacity of LNP-encapsulated ABE (adenine base editor), mRNA, and sgRNA_hP01, injected intravenously into C57BL/6J mice and cynomolgus monkeys. They were targeting the PCSK9 gene in the liver. They used the LNPs from the US 2016/0376224 A1 patent. Their LNPs had sizes between 67 and 71 nm. After two tail vein injections of 3 mg/kg total RNA per mouse, they obtained 67% base editing and a significant reduction in plasma PCSK9 and low-density lipoproteins (LDL). In cynomolgus monkeys, they obtained 28% base editing after one intravenous injection, which led to a 26% reduction in serum PCSK9 and a 9% reduction in serum LDL. After two injections, the base editing percentage remained unchanged, but they now observed a 39% of reduction in serum PCSK9 and a 19% reduction in serum LDL. They also studied the specificity of their LNP and concluded that it was indeed specific to the liver. Since this experiment was conducted on non-human primates, it is an important result for clinical applications.

Cheng et al. [94] developed the selective organ targeting (SORT) strategy. They found that the addition of the SORT molecule 1,2-dioleoyl-3-dimethylammonium-propane (DODAP) enhanced liver delivery. DODAP is an ionizable cationic SORT lipid with tertiary amino groups. To target the liver specifically, they found that the best LNP contained 20% DODAP. It had the following formulation: 5A2-SC8/DOPE/cholesterol/DMG-PEG/DODAP = 19.05/19.05/38.1/3.8/20 (mol/mol). The size of their LNPs was 155.1 nm. First, they intravenously injected these LNPs containing the Cas9 protein and a sgRNA targeting PTEN in a 2:1 ratio (w:w). They succeeded in editing PTEN exclusively in the liver with 13.9% gene editing. Secondly, they intravenously injected these LNPs containing the Cas9 mRNA with an sgRNA targeting PCSK9 in a 4:1 ratio (w:w). After three doses, they observed ~60% gene editing at the PCSK9 locus of liver tissue. This resulted in ~100% PCSK9 protein reduction in liver tissue and serum. They also evaluated the in vivo toxicity using a higher dose than needed. The LNPs used did not alter kidney or liver function. Serum cytokines were also not altered, and no adverse signs of injury were detected in the tissues. To check which liver cells were targeted by this LNP, they delivered Cre mRNA. Following a single injection (0.3 mg/kg), they observed that ~93% of all hepatocytes in the liver were targeted.

## 6. Targeting the Heart

To date, no study seems to have delivered the components of the CRISPR system efficiently and specifically to the heart by means of LNPs. However, some researchers have achieved the delivery of RNA, DNA, or ASO (Table 5). In this section, we review the study conducted by Scalzo et al. [115], which showed the most promising results.

Scalzo et al. [115] intravenously injected 0.2 µg of pDNA encapsulated in LNPs into C57BL/6 mice. They obtained a transfection efficiency greater than 60% on day 2 and greater than 80% on day 4. The treated group had GFP expression in the heart tissue that was two times higher than that of the control group. The formulation of their best LNP, named LNP4, was the following: C12-200/DOPE/cholesterol/DMG-PEG (35/56.5/6/2.5). The ionizable lipid:pDNA ratio was 10:1 (w:w). Their LNPs had a size of 114.7 nm. They also tested the safety of their formulations. After LNP treatment, they were no signs of an immune response in the heart. They also demonstrated that this LNP treatment did not affect cardiac cell function.

## 7. Targeting the Spleen

To date, only a few studies have delivered treatments specifically to the spleen with LNPs (Table 6). However, Cheng et al. [94] have successfully delivered the CRISPR-Cas9 components to the spleen.

Cheng et al. [94] used their SORT strategy. They found that the addition of the SORT molecule 1,2-dioleoyl-sn-glycero-3-phosphate (18PA) enhanced spleen delivery. To target the spleen specifically, they found that the best LNP contained 30% 18PA. It had the following formulation: 5A2-SC8/DOPE/cholesterol/DMG-PEG/18PA = 16.7/16.7/33.3/3.3/30 (mol/mol). They intravenously injected these LNPs containing the Cas9 protein and a sgRNA targeting PTEN in a 2:1 ratio (w:w) and observed gene editing. The size of their LNPs was 142.1 nm. To check which populations of spleen cells were targeted by this LNP, they delivered Cre mRNA. Following a single injection (0.3 mg/kg), they observed that these LNPs targeted 12% of B cells, 10% of T cells, and 20% of macrophages.

Sago et al. [99] developed a good LNP formulation to deliver the Cas9 mRNA and an sgRNA to the spleen epithelial cells. Their LNPs were composed of 7C1/DOPE/cholesterol/DMG-PEG (60/5/10/25). When delivering two doses of 2 mg total mRNA/kg of Cas9 mRNA and e-sgICAM2 (1:1 w:w), they obtained more than 20% indels.

## 8. Targeting the Bones

Many diseases can affect the bones. In most cases, they lead to disastrous consequences in the body. Glass bone disease (osteogenesis imperfecta) [117], Paget’s disease [118], and fibrous bone dysplasia [119] are just some examples. Osteoporosis is one of the most common bone diseases [120].

Few studies have attempted to target bone with LNPs [121]. Basha et al. [122] are among those who have obtained good results. They succeeded in delivering siSOST and siLuciferase via IV injections of LNPs. Their LNP formulation was the following: D-Lin-MC3-DMA/DSPC/cholesterol/DMG-PEG (50/10/38.5/1.5). Their ionizable lipid:RNA ratio was 10:1 (w:w). The size of their LNPs was 36.93 nm for siSOST particles and 37.88 nm for siLuciferase particles. They observed that about 50% of the osteocytes took up a significant amount of LNP-siRNA. SOST mRNA was also highly expressed in mouse bones. Seven days after the treatment, they observed 60% silencing relative to PBS and/or siCtrl. To support the hypothesis of the high specificity of these LNPs, they observed that SOST mRNA was minimally expressed in the kidneys, lungs, and spleen, and that only traces of SOST mRNA were found in the liver and heart.

## 9. Perspectives

In this review, we described the best existing formulations to target the muscles, brain, lungs, liver, heart, spleen, and bones. With the growing popularity of LNPs, more specific formulations will surely be published in the near future. The basic components of a good LNP have already been found. Now, small modifications remain to be made to increase the specificity of the delivery. Combining two improvements from two different studies will ensure that formulations can become more and more specific. Decorating LNPs with peptides is also a promising idea that can enhance specific organ targeting. Incorporating cell-membrane-derived components into LNPs is also a promising method to increase the specificity of LNPs [123]. In conclusion, this review will help researchers to select the best LNP formulations for future experiments.

## Figures and Tables

**Figure 1 pharmaceutics-14-02129-f001:**
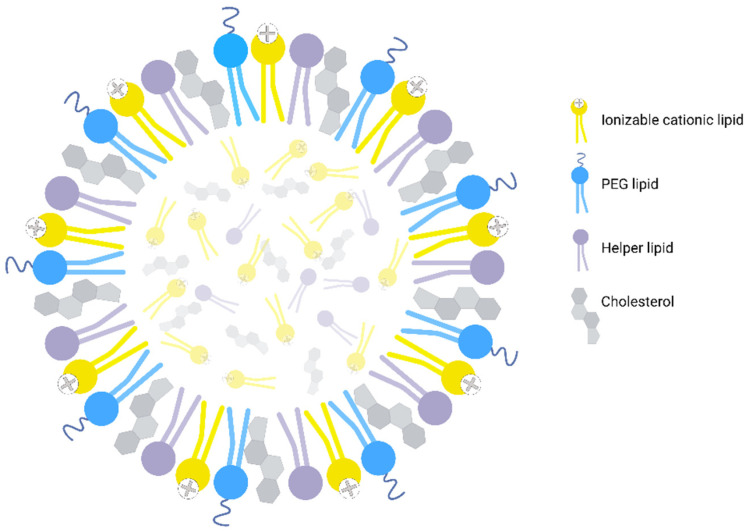
Key components of lipid nanoparticles (LNPs). LNPs are mainly composed of ionizable cationic lipids, helper lipids, cholesterol, and polyethylene glycol (PEG) lipids. Image created with BioRender.com.

**Figure 2 pharmaceutics-14-02129-f002:**
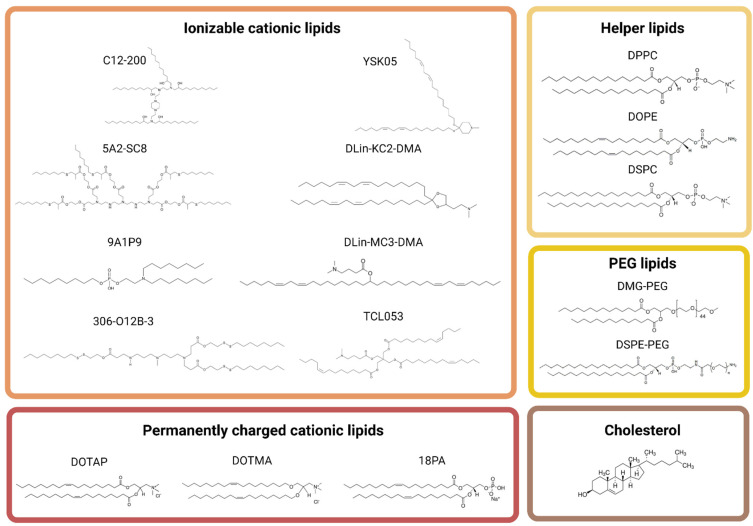
Chemical structures of some lipids used in LNPs. LNPs are generally composed of four categories of lipids: ionizable cationic lipids, helper lipids (phospholipids), cholesterol, and PEG lipids. A permanently charged cationic lipid can sometimes be added. TCL053, C12-200, YSK05, 5A2-SC8, DLin-K2-DMA, 9A1P9, DLin-MC3-DMA, and 306-O12B-3 are the principal ionizable cationic lipids used. DPPC, DOPE, and DSPC are the main helper lipids used. DMG-PEG and DSPE-PEG are the most used PEG lipids. There are three kinds of permanently charged cationic lipids that are mainly used: DOTAP, DOTMA, and 18PA. Image created with BioRender.com. The figure was adapted from [38,39,40,41,42,43,44,45,46,47,48,49,50,51,52,53,54].

**Table 1 pharmaceutics-14-02129-t001:** LNP formulations targeting muscles in vivo.

Reference	Delivered Cargo	Targeted Tissue/Gene	Route	No. of Doses	Dose	LNP Formulation	Lipids Molar Ratios	Lipid: RNA (or DNA) Ratio (w:w)	Results
	CRISPR-Cas9 mRNA/sgRNA		IV	1	1.0–10.0 mg/kg total RNA			23:1	Restoration of dystrophin
Kenjo 2021 [63]	CRISPR-Cas9 mRNA/sgRNA	Muscles /DMD	IM	1	10 μg Cas9 mRNA (with 10 μg sgRNA)	TCL053	60		∼10% exon skipping and ∼1.1% dystrophin recovery
				2		DPPC	10.6		∼13% exon skipping and ∼2.6% dystrophin recovery
				3		Cholesterol	27.3		∼15% exon skipping and ∼4.0% dystrophin recovery
				6		DMG-PEG	2.1		Restoration of dystrophin in 38.5% of muscle fibers
Wei 2020 [66]	RNP	Muscles /DMD	IM	1	1 mg/kg sgRNA	5A2-SC8	21.4	40:1	td-Tom fluorescence near the injection site
						DOPE	21.4		
						Cholesterol	42.8		
				3		DMG-PEG	4.3		4.2% restoration of dystrophin
						DOTAP	10		
Guimaraes 2019 [67]	b-mRNA	Muscles	IV	1	0.25 μg de b-mRNA	C12–200	35	5:1 ^1^	
						DOPE	16		
						Cholesterol	46.5	7.5:1 ^1^	
						DMG-PEG	2.5		
Carrasco 2021 [65]	FLuc mRNA	Muscles	IM		5 μg	DLin-KC2-DMA	50	4:1 ^1^ (mol:mol)	
						DSPC	10		
						Cholesterol	38.5		
						DMG-PEG	1.5		
Blakney 2019 [64]	FLuc saRNA	Muscles	IM		5 μg	C12-200	35	12:1 ^1^	
						DOPE	16		
						Cholesterol	49		

^1^ Ionizable lipid:RNA ratio. 5A2-SC8 = C_93_H_173_N_5_O_20_S_5_; ASO = antisense oligonucleotide; C12-200 = 1,1′-[[2-[4-[2-[[2-[bis(2-hydroxydodecyl)amino]ethyl](2-hydroxydodecyl)amino]ethyl]-1-piperazinyl]ethyl]imino]bis-2-dodecanol; DLin-KC2-DMA = 2,2-dilinoleyl-4-(2-dimethylaminoethyl)-[1,3]-dioxolane; DMD = dystrophin gene; DOPE = 1,2-dioleoyl-sn-glycero-3-phosphoethanolamine; DOTAP = 1,2-dioleoyl-3-trimethylammonium-propane; DPPC = 1,2-dipalmitoyl-sn-glycero-3-phosphocholine; DSPC = distearoylphosphatidylcholine; FLuc = firefly luciferase; IM = intramuscular; IP = intraperitoneal; IV = intravenous; PEG3000-C16 = 1,2-dipalmitoyl-sn-glycero-3-phosphoethanolamine-N-[methoxy(polyethylene glycol)-3000]; DMG-PEG = 1,2-Dimyristoyl-rac-glycero-3-methoxypolyethylene glycol-2000; RNP = Ribonucleoproteins; TCL053 = 2-(((4-(dimethylamino)butanoyl)oxy)methyl)-2-((((Z)-tetradec-9-enoyl)oxy)methyl)propane-1,3-diyl (9Z,9’Z)-bis(tetradec-9-enoate); w:w = weight:weight.

**Table 2 pharmaceutics-14-02129-t002:** LNP formulations targeting the brain in vivo.

Reference	Delivered Cargo	Targeted Tissue/Gene	Route	No. of Doses	Dose	LNP Formulation	Lipids Molar Ratios	Lipid: RNA (or DNA) Ratio (w:w)	Results
Ma 2020 [88]	ASO targeting tau mRNA	Brain/tau	IV	5	1 mg/kg	306-O12B-3	67.2 (w)		~50% reduction of tau mRNA and ~30% reduction of tau protein
						DSPE-PEG	4 (w)		
						NT1-O14B	28.8 (w)		
	(-27)GFP-Cre protein	Brain	IV	4	50 μg per injection	PBA-Q76-O16B	67.2 (w)		Strong tdTomato signals were observed in multiple regions of the brain, including the cerebral cortex, hippocampus, and cerebellum.
						DSPE-PEG	4 (w)		
						NT1-O14B	28.8 (w)		
Wei 2020 [66]	RNP	Brain	IC	1	0.15 mg/kg sgRNA	5A2-SC8	21.4	40:1	
						DOPE	21.4		
						Cholesterol	42.8		
						DMG-PEG	4.3		
						DOTAP	10		
Nabhan 2016 [89]	mRNA	DRG/FXN	ICV can	1	0.2 mg/kg	DLin-MC3-DMA DSPC Cholesterol DMG-PEG	55	30:1	Robust increase in mFXN levels.
							10		
			IT				32.5		LNP-derived human mFXN levels were ~3-fold higher than mouse mFXN in the control group
							2.5		
Tamaru 2014 [90]	DNA encoding mCherry	BECs	ICV			YSK05	70		
						Cholesterol	30		
						DMG-PEG	3		

5A2-SC8 = C_93_H_173_N_5_O_20_S_5_; ASO = antisense oligonucleotide; BECs = brain endothelial cells; C12-200 = 1,1′-[[2-[4-[2-[[2-[bis(2-hydroxydodecyl)amino]ethyl](2-hydroxydodecyl)amino]ethyl]-1-piperazinyl]ethyl]imino]bis-2-dodecanol; Cre = Cre recombinase is a tyrosine recombinase enzyme derived from the P1 bacteriophage; DLin-KC2-DMA = 2,2-dilinoleyl-4-(2-dimethylaminoethyl)-[1,3]-dioxolane; DOPE = 1,2-dioleoyl-sn-glycero-3-phosphoethanolamine; DOTAP = 1,2-dioleoyl-3-trimethylammonium-propane; DRG = dorsal root ganglia; DSPC = distearoylphosphatidylcholine; FLuc = firefly luciferase; FXN = frataxin gene; GFP = green fluorescent protein; ICV can = intra-cerebroventricular cannulae; IC = intracranial; IT = intrathecal; IV = intravenous; DMG-PEG1000-C18 = 1,2-distearoyl-sn-glycero-3-phosphoethanolamine-N-[methoxy(polyethylene glycol)-1000]; DMG-PEG = 1,2-dimyristoyl-rac-glycero-3-methoxypolyethylene glycol-2000; RNP = ribonucleoproteins; tau = tubulin-associated unit; w = weight; w:w = weight:weight; YSK05 = 1-methyl-4,4-bis [9Z,12Z]-ocatadeca-9,12-dien-1-yloxy]piperidine.

**Table 3 pharmaceutics-14-02129-t003:** LNP formulations targeting the lungs in vivo.

Reference	Delivered Cargo	Targeted Tissue/Gene	Route	No. of Doses	Dose	LNP Formulation	Lipids Molar Ratios	Lipid: RNA (or DNA) Ratio (w:w)	Results
Cheng 2020 [94]	Cas9 protein + sgRNA (RNP)	Lungs/PTEN	IV	1	1.5 mg/kg sgRNA	5A2-SC8	11.9	40:1	5.3% gene editing
						DOPE	11.9		
						Cholesterol	23.8		
	Cas9 mRNA/sgRNA				2.5 mg/kg total RNA	DMG-PEG	2.4		15.1% gene editing
						DOTAP	50		
Wei 2020 [66]	RNP	Lungs PTEN	IV	1	1.5 mg/kg sgRNA	5A2-SC8	11.9	40:1	13% indel frequency
		P53; PTEN; EMl4; ALK; RB1		1	0.33 mg/kg each sgRNA	DOPE	11.9		Gene editing: 1.1% (P53) 3.4% (PTEN); 7.7% (EMl4); 1.1% (ALK); 7.5% (RB1)
						Cholesterol	23.8		
		Eml4/Alk		1	2 mg/kg sgRNA	DMG-PEG	2.4		16.3% (EMl4); 4.5% (ALK)
		Eml4/Alk		2	1.5 mg/kg sgRNA	DOTAP	50		13.1% (EMl4); 3.5% (ALK)
Robinson 2018 [95]	cmCFTR mRNA	Lungs/CFTR	IN		0.6 mg cmRNA/kg	DLin-MC3-DMA	50		Polarization in response to CFTR
						DSPC	10		
	cmFLuc mRNA					Cholesterol	38.5		
						DMG-PEG	1.5		
Zhang 2020 [96]	FLuc mRNA	Heart, Spleen, Lung	IV		0.5 mg/kg	FTT7 lipids	22.04		
						DOPE	33.06		
						Cholesterol	44.08		
						DMG-PEG	0.82		
Paunovska 2018 [97]		Lung EC	IV			7C1	62		
						Cholesteryl Stearate	30		
						DMG-PEG	8		
		Lung Macs				7C1	50		
						DOPE	8		
						7B-OH Cholesterol	40		
						DMG-PEG	2		
Paunovska 2018 [98]	b-DNA	Heart ECs, Lung ECs	IV		0.75 mg/kg	104-PEI600	62		
						Lipid = C12Epoxy	5		
						PEG350-C18	33		
		Lungs Macs				7C1-PEI600	62		
						Lipid = C15Epox	5		
						PEG350-C14	33		
		Lungs Ecs				7C1-PEI600	62		
						Lipid = C15Epoxy	6		
						DMG-PEG	32		
		Lung Macs				104-PEI600	62		
						Lipid = C12Epoxy	21		
						DMG-PEG-C18	17		
		Heart Macs and Lungs Macs				104-PEI600	62		
						Lipid=C12Epoxy	18		
						DMG-PEG-C18	20		
Sago 2018 [99]	two sgRNAs targeting ICAM2 (sgICAM2ab	Lung, Spleen, Kidney	IV	3	1.5 mg/kg	7C1	50		Good for small RNAs but not for mRNAs
						18:1Lyso PC	20		
						Cholesterol	23.5		
						DMG-PEG	6.5		
Qiu 2022 [100]	Cas9 mRNA + sgRNA	Lungs/LoxP	IV		1.67 mg/kg	306-N16B	50	10:1^1^	
						DOPE (or DSPC)	10		
						Cholesterol	38.5		
						DMG-PEG	1.5		
Liu 2021 [101]	Cre mRNA	Lungs	IV			9A1P9	46		Transfection of ~34% of all endothelial cells, ~20% of all epithelial cells, and ~13% of immune cells
						DDAB	23		
	Cas9 mRNA + Tom1 sgRNA	Lungs			0.75 mg/kg	Cholesterol	30.7		Specific gene editing in the lungs
	Cas9 mRNA + sgRNA	PTEN			0.75 mg/kg	DMG-PEG	0.3		Efficient target gene editing
Hagino 2021 [102]	pDNA + PEI	Lungs	IV		40 μg de pDNA	The inner coat (half of the total lipid): DOPE STR-R8		640 nmol of lipid for 30 μg de pDNA	High gene Expression in the lungs
							9.55		
							0.45		
						The outer coat: DOTMA YSK05 Cholesterol DMG-PEG Chol-GALA			
							4		
							4		
							2		
							0.3		
							0.4		

^1^ Active lipidoid:mRNA ratio. 5A2-SC8 = C_93_H_173_N_5_O_20_S_5_; ALK = anaplastic lymphoma kinase gene; C12-200 = 1,1′-[[2-[4-[2-[[2-[bis(2-hydroxydodecyl)amino]ethyl](2-hydroxydodecyl)amino]ethyl]-1-piperazinyl]ethyl]imino]bis-2-dodecanol; CFTR = cystic fibrosis transmembrane conductance regulator gene; Cre = Cre recombinase is a tyrosine recombinase enzyme derived from the P1 bacteriophage; DDAB = dimethyldioctadecylammonium; DLin-KC2-DMA = 2,2-dilinoleyl-4-(2-dimethylaminoethyl)-[1,3]-dioxolane; DOPE = 1,2-dioleoyl-sn-glycero-3-phosphoethanolamine; DOTAP = 1,2-dioleoyl-3-trimethylammonium-propane; DSPC = distearoylphosphatidylcholine; ECs = epithelial cells; FLuc = firefly luciferase; GFP = green fluorescent protein; ICAM2 = intercellular adhesion molecule 2 gene; IN = intranasal; IV = intravenous; Macs = macrophages; P53 = tumor protein 53; DMG-PEG1000-C18 = 1,2-distearoyl-sn-glycero-3-phosphoethanolamine-N-[methoxy(polyethylene glycol)-1000]; DMG-PEG = 1,2-dimyristoyl-rac-glycero-3-methoxypolyethylene glycol-2000; PEI = cationic polymer polyethyleneimine; PTEN = phosphatase and tensin homolog; RB1 = retinoblastoma 1 gene; RNP = ribonucleoproteins; w:w = weight:weight; YSK05 = 1-methyl-4,4-bis [9Z,12Z]-ocatadeca-9,12-dien-1-yloxy]piperidine.

**Table 4 pharmaceutics-14-02129-t004:** LNP formulations targeting the liver in vivo.

Reference	Delivered Cargo	Targeted Tissue/Gene	Route	No. of Doses	Dose	LNP Formulation	Lipids Molar Ratios	Lipid: RNA (or DNA) Ratio (w:w)	Results
Cheng 2020 [94]	Cas9 mRNA/sgRNA	Liver/PTEN	IV	1	2.5 mg/kg total RNA	5A2-SC8	19.05	40:1	2.7% gene editing
	Cas9 protein + sgRNA (RNP)			1	1.5 mg/kg sgRNA	DOPE	19.05		11.6%–13.9% gene editing
						Cholesterol	38.1		
	Cas9 mRNA/sgRNA	Liver/PCSK9		3	2.5 mg/kg total RNA	DMG-PEG	3.8		∼60% gene editing 100% reduction in liver and serum Pcsk9
						DOTAP	20		
Wei 2020 [66]	RNP	Liver/P53, PTEN, RB1	IV	3	2.5 mg/kg sgRNA	5A2-SC8	22.6	40:1	8.6% (P53); 7.9% (PTEN); 13.3% (RB1) gene editing
						DOPE	22.6		
						Cholesterol	45.2		
		Liver/PCSK9				DMG-PEG	4.5		5.7% gene editing reduction of PCSK9 in serum and liver tissue
						DOTAP	5		
Zhang 2020 [96]	hFVIII mRNA	Liver	IV		2 mg/kg	FTT5 lipids	22.04		Restores the hFVIII level up to 90% of normal activity.
						DOPE	33.06		
	mRNA encoding ABE + sgRNA	PCSK9			1 mg/kg of total RNA dose	Cholesterol	44.08		60% of gene editing
						DMG-PEG	0.82		
Paunovska 2018 [97]		Liver EC	IV			7C1	50		
						18:1Lyso PC	10		
						4B-OH-Cholesterol	29		
						DMG-PEG	11		
		Liver hepatocyte				7C1	50		
						DOPE	8		
						7B-OH Cholesterol	40		
						DMG-PEG	2		
Cui 2022 [111,112]	FLuc mRNA	Liver; iWAT, gWAT	IV		0.25 mg/kg	MC3	50	20:1	
						DSPC	10		
						Cholesterol	38.5		
						DMG-PEG	1.5		
Rothgangl 2021 [113]	ABE + gRNA	Liver/PCSK9	IV	1	1.0, 1.5, or 3.0 mg/kg total RNA	Patent: US 2016/0376224 A1			In mice: 4%, 12%, or 51% base editing
				2	1.5 or 3.0 mg/kg total RNA				In mice: 40% or 67% base editing
				1	1.5 mg/kg total RNA				In cynomolgus monkeys: 28% base editing 26% reduction in serum PCSK9
				2	1.5 mg/kg total RNA				In cynomolgus monkeys: 24% base editing 39% reduction in serum PCSK9
Liu 2019 [22]	Cas9 mRNA + gRNA	Liver, Kidney/PCSK9	IV	1	0.6 mg/kg Cas9 mRNA	BAMEA-16B	16 (w)	15:1	Reduction of serum PCSK9 by 80%
						DOPE	4 (w)		
						Cholesterol	8 (w)		
						DSPE-PEG	1 (w)		
Prakash 2013 [114]	ASO	Liver/PTEN	IV	1	4.5 mg/kg	DLin-KC2-DMA	57.5	10:1	In mice: around 85% of reduction of PTEN mRNA
						DSPC	7.5		
	ss-siRNA					Cholesterol	31.5		In mice: around 70% of re-duction of PTEN mRNA
	siRNA					DMG-PEG	3.5		In mice: around 75% of re-duction of PTEN mRNA

5A2-SC8 = C_93_H_173_N_5_O_20_S_5_; ABE = adenine base editor; ALK = anaplastic lymphoma kinase gene; C12-200 = 1,1′-[[2-[4-[2-[[2-[bis(2-hydroxydodecyl)amino]ethyl](2-hydroxydodecyl)amino]ethyl]-1-piperazinyl]ethyl]imino]bis-2-dodecanol; DLin-KC2-DMA = 2,2-dilinoleyl-4-(2-dimethylaminoethyl)-[1,3]-dioxolane; DOPE = 1,2-Dioleoyl-sn-glycero-3-phosphoethanolamine; DOTAP = 1,2-dioleoyl-3-trimethylammonium-propane; DSPC = distearoylphosphatidylcholine; ECs = epithelial cells; FLuc = firefly luciferase; GFP = green fluorescent protein; gWAT = gonadal white adipose tissue; hFVIII = human factor VIII; IV = intravenous; iWAT = inguinal white adipose tissue; Macs = macrophages; P53 = tumor protein 53; PEG1000-C18 = 1,2-distearoyl-sn-glycero-3-phosphoethanolamine-N-[methoxy(polyethylene glycol)-1000]; DMG-PEG = 1,2-dimyristoyl-rac-glycero-3-methoxypolyethylene glycol-2000; PTEN = phosphatase and tensin homolog; RB1 = retinoblastoma 1 gene; RNP = ribonucleoproteins; w = weight; w:w = weight:weight.

**Table 5 pharmaceutics-14-02129-t005:** LNP formulations targeting the heart in vivo.

Reference	Delivered Cargo	Targeted Tissue/Gene	Route	No. of Doses	Dose	LNP Formulation	Lipids Molar Ratios	Ionizable Lipid: RNA (or DNA) Ratio (w:w)	Results
Scalzo 2022 [115]	pDNA	Cardiac cells	IV		0.2 µg of pDNA	C12-200	35	10:1	Transfection efficiency
						DOPE	56.5		Day 2 = 60%
						Cholesterol	6		Day 4 = 80%
						DMG-PEG	2.5		A twofold increase in GFP expression in the heart tissue compared to the control group
Zhang 2020 [96]	FLuc mRNA	Heart, Spleen, Lung	IV		0.5 mg/kg	FTT7 lipids	22.04		
						DOPE	33.06		
						Cholesterol	44.08		
						DMG-PEG	0.82		
Paunovska 2018 [98]	b-DNA	Heart ECs, Lung ECs	IV		0.75 mg/kg	104-PEI600	62		
						Lipid=C12Epoxy	5		
						DMG-PEG350-C18	33		
		Heart ECs and Macs				102-Spermidine	35		
						Lipid=C12Epoxy	35		
						DMG-PEG	30		
		Heart Ecs				104-PEI600	62		
						DMG-PEG-C18	38		
		Heart Macs and Lungs Macs				104-PEI600	62		
						Lipid=C12Epoxy	18		
						DMG-PEG-C18	20		

ASO = antisense oligonucleotide; C12-200 = 1,1′-[[2-[4-[2-[[2-[bis(2-hydroxydodecyl)amino]ethyl](2-hydroxydodecyl)amino]ethyl]-1-piperazinyl]ethyl]imino]bis-2-dodecanol; DOPE = 1,2-dioleoyl-sn-glycero-3-phosphoethanolamine; ECs = epithelial cells; FLuc = firefly luciferase; GFP = green fluorescent protein; IV = intravenous; Macs = macrophages; DMG-PEG = 1,2-dimyristoyl-rac-glycero-3-methoxypolyethylene glycol-2000; w:w = weight:weight.

**Table 6 pharmaceutics-14-02129-t006:** LNP formulations targeting the spleen in vivo.

Reference	Delivered Cargo	Targeted Tissue/Gene	Route	No. of Doses	Dose	LNP Formulation	Lipids Molar Ratios	Ionizable Lipid: RNA (or DNA) Ratio (w:w)	Results
Cheng 2020 [94]	Cas9 mRNA/sgRNA	Spleen/PTEN	IV	1	4.0 mg/kg	5A2-SC8	16.7	40:1 (total lipid: mRNA)	
						DOPE	16.7		
						Cholesterol	33.3		
						DMG-PEG	3.3		
						18PA	30		
Zhang 2020 [96]	FLuc mRNA	Spleen	IV		0.5 mg/kg	FTT3 lipids	22.04		
						DOPE	33.06		
						Cholesterol	44.08		
						DMG-PEG	0.82		
Paunovska 2018 [97]		Spleen EC	IV			7C1	50		
						DSPC	8		
						Cholesteryl Stearate	40		
						DMG-PEG	2		
		Spleen Macs				7C1	50		
						DOPE	8		
						7B-OH Cholesterol	40		
						DMG-PEG	2		
Sago 2018 [99]	SpCas9 mRNA and sgICAM2ab	Spleen ECs	IV	2	2 mg/kg	7C1	60		The best formulations for mRNA
						DOPE	5		
						Cholesterol	10		
	SpCas9 mRNA + e-sgICAM2					DMG-PEG	25		
	two sgRNAs targeting ICAM2 (sgICAM2ab)	Lung, Spleen, Kidney		3	1.5 mg/kg	7C1	50		Good for small RNAs but not for mRNA
						18:1Lyso PC	20		
						Cholesterol	23.5		
						DMG-PEG	6.5		
Maugeri 2019 [116]	hEPO mRNA	Spleen	IV		1.5 µg per mouse	DLin-MC3-DMA	50	3:1	
						DSPC	10		
						Cholesterol	38.5		
						DMPE-PEG	1.5		

18PA = 1,2-dioleoyl-sn-glycero-3-phosphate; 5A2-SC8 = C_93_H_173_N_5_O_20_S_5_; C12-200 = 1,1′-[[2-[4-[2-[[2-[bis(2-hydroxydodecyl)amino]ethyl](2-hydroxydodecyl)amino]ethyl]-1-piperazinyl]ethyl]imino]bis-2-dodecanol; DLin-KC2-DMA = 2,2-dilinoleyl-4-(2-dimethylaminoethyl)-[1,3]-dioxolane; DOPE = 1,2-dioleoyl-sn-glycero-3-phosphoethanolamine; DSPC = distearoylphosphatidylcholine; ECs = epithelial cells; FLuc = firefly luciferase; GFP = green fluorescent protein; hEPO = human erythropoietin; ICAM2 = intercellular adhesion molecule 2 gene; IV = intravenous; Macs = macrophages; DMG-PEG = 1,2-dimyristoyl-rac-glycero-3-methoxypolyethylene glycol-2000; PTEN = phosphatase and tensin homolog; RNP = ribonucleoproteins; w:w = weight:weight.

## Data Availability

Not applicable.

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
