# Peer review of "Delivery of RNAs to Specific Organs by Lipid Nanoparticles for Gene Therapy"

_pharmaceutics, 2022, doi:10.3390/pharmaceutics14102129_

Round 1

Reviewer 1 Report (Previous Reviewer 1)

This is a resubmission of a paper that I already considered suitable for publication. I notice the authors have removed the figures prepared from their calculations using the raw data obtained by the work of Dahlman et al, as well as the pancreas targeting section. These corrections do not change my positive evaluation of the work, that can be published in its present form, following:

1. removal of the word "pancreas" from line 404 (it is not discussed in this work anymore)

2. revising the manuscript for an optimal use of english, avoiding unnecessary repetitions (e.g. line 409 "great promising idea")

Author Response

Reviewer 2 Report (New Reviewer)

Manuscript Number: Pharmaceutics-1934262

Manuscript Title: Delivery of RNAs to Specific Organs by Lipid Nanoparticles  for Gene Therapy

Additional Comments:

Lipid nanoparticles-based delivery method have demonstrated its value in developing gene therapy and antisense therapy.  But its application is limited to liver specific indications. Developing LNP technologies for extrahepatic tissues are vital for developing gene therapy and oligonucleotide therapeutics. In this review, authors summarized the most effective compositions and proportions of lipids for LNPs to target specific organs, such as the brain, lungs, muscles, heart, liver, spleen, and bones. This review is well written, and most of the relevant references are cited. In my opinion, this article is suitable for publication in Pharmaceutics after minor revision.

One of the early publications on LNP delivery of ASO, siRNA and ss-sRNA should be cited in this review

“Lipid Nanoparticles Improve Activity of Single-Stranded siRNA and Gapmer Antisense Oligonucleotides in Animals. ACS Chemical Biology 2013, 8, 1402-1406”

Author Response

Reviewer 3 Report (New Reviewer)

In this manuscript, the authors present an interesting and well-written review about the delivery of nucleic acids to specific organs through lipid nanoparticles. The paper describes different publications in which the composition of LNP intended for targeting to specific organs is related to efficacy; however, the involved mechanism for targeting is not discussed. Moreover, nothing is said about the particle size and surface characteristics of the nanoparticles, which are also very important for biodistribution.

Minor comments that may help the authors to improve the manuscript:

-        Lines 52-56. Regarding ionizable lipids, the authors should mention their mechanism of action (the ability of ionizable lipids to form destabilizing nonbilayer structures at acidic pH is recognized as the key for endosomal escape and, therefore, cytosolic delivery).

-        Lines 290-291. Among the objectives to transfect the liver, consider this organ as a protein factory.

-        Lines 354-361. In the work of Scalzo, LNPs were developed for pDNA delivery in cardiomyocytes; however, in line 359, authors describe a lipid:mRNA ratio.

Round 2

Reviewer 3 Report (New Reviewer)

The author response in not convincing. It is true that, in general, it is difficult to relate the physicochemical characteristics of the nanoparticles to the biodistribution; however, some studies have related the uptake of the nanoparticles by specific organs with the size or the shape. For instance, non-spherical nanocarriers have shown enhanced splenic delivery of active agents by avoiding hepatic uptake. Hepatocytes internalization is also conditioned by the particle size (in humans, less than 100 nm). On the other hand, the functionalization of the systems with specific ligands helps to target to specific tissues (for instance monoclonal antibodies). These comments should be discussed in the manuscript. In my opinion, and according to the title, this information is expected to be found in the manuscript.

Author Response

This manuscript is a resubmission of an earlier submission. The following is a list of the peer review reports and author responses from that submission.

Round 1

Reviewer 1 Report

This review systematically analyzes the literature discussing the composition of LNPs and the impact on organ targeting. The analysis of literature is sufficiently broad and the structure is well organized. The topic is timely and of interest for the readers of Pharmaceutics. However, some corrections should be made before publishing, including deleting some unnecessary repetitions, improving the description of some papers, clarifying certain comments and including some important missing documents. See below a list of suggestions.

I wish to congratulate with the authors for the comprehensive and well-structured tables. However, as the focus of the review is the composition of LNPs, I suggest adding one figure that reports the chemical structure of each lipid mentioned in the tables, or at least of each ionizable/cationic lipid.

Also about the tables: please double check the lipid molar ratios. For some of the references (e.g. 48, 79, 84) the sum of molar ratios is not 100.

Please make the nomenclature of lipids uniform: this would help the reader to immediately spot differences and similarities between formulations reported by different groups. For example, DMG-PEG is sometimes referred to as C14-PEG, or C14-PEG2000, or DMG-PEG2000, or PEG-DMG, or PEG2000-C14, or PEG2000C14: I suggest to choose one of these, to define it in the legend providing the iupac name, and to use it consistently in the text and the tables.

I would remove the unnecessary general comments without appropriate context (e.g. line 128: The study of Ferlini et al. is very interesting; line 248: They obtained excellent results).

Note that the nanoparticles reported in Ferlini et al (reference 51) are not LNPs and do not belong to this review. They are in fact polymeric nanoparticles. If you consider this study of particular interest for its biologic results, please explicitly state in the text that these are not LNPs (a sentence like “Despite not being LNPs, the polymeric vector described by Ferlini et al demonstrated interesting efficacy data in vivo, that we decided to include in this review because…” would work).

The sentence in line 143-144 might be misleading. When you state “AONs have since been approved”, do you mean that the same AON used by Ferlini has been clinically approve? Or you refer to AONs as a class of molecules? Please specify.

The sentence in line 145 is strangely composed. It implies that works not using CRISPRs are generally not providing useful results, that is obviously not true. Moreover, in the previous paragraph you describe an exon skipping approach, that is not CRISPR as well. I would remove the sentence.

Line 146: the acronym LNPs have already been defined, no need to explicit “lipid nanoparticles” again.

Regarding the calculations included in the Supplementary File 1: providing the raw data is appreciable, but I suggest to include a short paragraph (maybe in the first sheet of the excel file) describing the methods you used for these calculations. Also, please add to the caption of the figures if the % is intended as % of the injected dose, or if it is normalized by organ weight.

Line 180-182: I appreciate the concluding remarks at the end of the section, but these is no need to repeat the lipid composition that is already reported before (and in the table). About the final part of the concluding remarks (lines 189-193): from your description, every study seems promising and worth of further investigation, but this statement is quite general and do not really add much to the discussion.

Reference 71 is reported in a wrong format in the reference list.

Line 220: I understand the excitement about the result, but exclamation marks should be avoided.

The text in lines 84-103 and lines 293-310 is the same, reported twice. Repetition of the same concept should be avoided unless necessary and -in any case- you cannot repeat the same identical text.

When talking of the liver targeting of LNPs, it would be important to highlight the role of Apoliporotein E, that forms a corona around the particles upon IV injection and directs them towards hepatocytes. This feature is particularly relevant as it allowed a high delivery and optimal biological effect of Onpattro, the first siRNA-LNPs approved (see doi: 10.1038/mt.2010.85 and doi: 10.1038/s41565-019-0591-y).

Regarding the LNPs targeting the heart: please check if the recent paper from Evers et al (doi: 10.1016/j.jconrel.2022.01.027) should be included.

Some minor revisions of English are required (e.g. line 56: Duchesne's/Duchenne’s; line 64: to targeted/to target; line 125: prevent/present; line 161: one of the only/one of the few; line 235 and 239: to delivered/to deliver; line 293: to encapsulated/to encapsulate etc etc)

Author Response

See the Word file.

Reviewer 2 Report

Lipid nanoparticles (LNPs) are very promising carriers for targeted delivery. They provide several advantages including high encapsulation efficiency, high stability, and low toxicity. In addition, they can be actively targeted to specific organs and cell types when conjugated to targeting moieties such as antibodies and peptides.

The authors made a good effort to gather and summarize their findings on LNPS targeted to different organs. However, this manuscript has several issues that are listed below.

1. There are several syntax and structural errors throughout the manuscript that makes it hard for the reader to focus on the scientific content. Several sections of this manuscript are poorly written and need a thorough revision. Below are only few examples of structural issues.

 ·         Line 64, few studies have succeeded to target instead of “a few studies had succeeded to targeted..”

·         Add the abbreviation DMD next to Duchenne muscular dystrophy in line 56 when it was first used.

·         Line 84, succeeded to encapsulate and not to encapsulated..and to deliver them and not to delivered them…

·         Line 87 Does not need to be translated instead of …translate…

·         Line 102, There LPNS were…?

·         Line 115: what do the authors mean by “permanent cationic lipids”?

·         Line 125, what do the authors mean by: However, “they did not prevent quantitative results”?

·         Line 180, their composition instead of “There composition”.

·         Line 195, what do the authors mean by ”we only have to think about Alzheimer’s disease…”?

·         Lines 293-294, Wei et al. [48] had an interesting composition, because they succeed to encapsulated 293 RNPs and to delivered them to the lungs by tail vein injection” example of poorly written sentence.

The manuscript needs an extensive editing in order to make it fit for publication.

 2. The authors need to include a section for all the abbreviations in the text. It is hard for the reader to follow the meaning of the text while having to go back searching what an abbreviation stands for.

3. The quality of the figures could be improved by using a better tool such as GraphPad.

4. The authors could include a section on actively targeted LNPs, conjugated with antibodies or peptides.

5. The tables could use some editing and formatting to make them easier to read and follow.

6. The manuscript could benefit from using one or two summary/comprehensive figures.

Author Response

See the Word file.

Reviewer 3 Report

Delivery of RNAs to Specific Organs by Lipid Nanoparticles for Gene Therapy.

Manuscript ID: pharmaceutics-1868134

The authors have discusses about the delivery of LNP’s to specific organs. In this review the authors have discussed the most effective compositions and pro- 15 portions of lipids for LNPs to target specific organs, such as the brain, lungs, muscles, heart, pancreas, liver, spleen, and bones. The manuscript requires major revisions:

Major comments:

1. The introduction and the discussion section should be improved.

2. Figures and pictorial representations of the of LNP’s should be included.

3. What are the factors that determine the Fate of LNP’s into various tissues? This should be discussed. What are the physiochemical properties of the LNP’s that affect this. This should be discussed.

4. The current figures in the manuscript should be taken directly from the referenced articles, and should not be plotted by the authors.

5. A section that discusses the current gene therapy in the clinic which target different tissues can be discussed.

Minor changes:

None. 

Author Response

See the Word file.

Round 2

Reviewer 1 Report

The authors significantly improved the review that can be published after the following important revisions are made:

1. Figure 2. Note that DOTMA is a permanently charged lipid, not ionizable lipid

2. Table 3, line referring to Sago et al: what is ARNm? Please define it among the acronyms or correct if misspelled

3. Sentence in line 779-780 is not entirely correct: Patisiran is the first RNA drug delivered by LNPs approved by FDA, but not the only one (now also the mRNA vaccines by Pfizer/Biontech have been approved).

4. In the supplementary material please correct efficacity with efficacy

Author Response

See file.

Reviewer 3 Report

The authors have answered all the concerns raised by the reviewer. The manuscript can be accepted in the current form. 

Author Response

Thank you for reviewing my manuscript!